# Microphysiological Neurovascular Barriers to Model the Inner Retinal Microvasculature

**DOI:** 10.3390/jpm12020148

**Published:** 2022-01-24

**Authors:** Thomas L. Maurissen, Georgios Pavlou, Colette Bichsel, Roberto Villaseñor, Roger D. Kamm, Héloïse Ragelle

**Affiliations:** 1Roche Pharma Research and Early Development, Immunology, Infectious Diseases and Ophthalmology, Roche Innovation Center Basel, F. Hoffmann-La Roche Ltd., Grenzacherstrasse 124, 4070 Basel, Switzerland; thomas.maurissen@roche.com; 2Department of Biological Engineering, Massachusetts Institute of Technology, 77 Massachusetts Ave., MIT Building, Room NE47-321, Cambridge, MA 02139, USA; gpavlou@mit.edu; 3Roche Pharma Research and Early Development, Pharmaceutical Sciences, Roche Innovation Center Basel, F. Hoffmann-La Roche Ltd., Grenzacherstrasse 124, 4070 Basel, Switzerland; colette.bichsel.cb1@roche.com; 4Roche Pharma Research and Early Development, Institute for Translational Bioengineering, Roche Innovation Center Basel, F. Hoffmann-La Roche Ltd., Grenzacherstrasse 124, 4070 Basel, Switzerland; 5Roche Pharma Research and Early Development, Neuroscience and Rare Diseases, Roche Innovation Center Basel, F. Hoffmann-La Roche Ltd., Grenzacherstrasse 124, 4070 Basel, Switzerland; roberto.villasenor_solorio@roche.com; 6Department of Mechanical Engineering, Massachusetts Institute of Technology, 77 Massachusetts Ave., MIT Building, Room NE47-321, Cambridge, MA 02139, USA

**Keywords:** microphysiological systems, blood-neural barriers, neurovascular unit, disease modeling, 3D models, organ-on-a-chip, inner blood-retinal barrier

## Abstract

Blood-neural barriers regulate nutrient supply to neuronal tissues and prevent neurotoxicity. In particular, the inner blood-retinal barrier (iBRB) and blood–brain barrier (BBB) share common origins in development, and similar morphology and function in adult tissue, while barrier breakdown and leakage of neurotoxic molecules can be accompanied by neurodegeneration. Therefore, pre-clinical research requires human in vitro models that elucidate pathophysiological mechanisms and support drug discovery, to add to animal in vivo modeling that poorly predict patient responses. Advanced cellular models such as microphysiological systems (MPS) recapitulate tissue organization and function in many organ-specific contexts, providing physiological relevance, potential for customization to different population groups, and scalability for drug screening purposes. While human-based MPS have been developed for tissues such as lung, gut, brain and tumors, few comprehensive models exist for ocular tissues and iBRB modeling. Recent BBB in vitro models using human cells of the neurovascular unit (NVU) showed physiological morphology and permeability values, and reproduced brain neurological disorder phenotypes that could be applicable to modeling the iBRB. Here, we describe similarities between iBRB and BBB properties, compare existing neurovascular barrier models, propose leverage of MPS-based strategies to develop new iBRB models, and explore potentials to personalize cellular inputs and improve pre-clinical testing.

## 1. Introduction

The inner blood-retinal barrier (iBRB) is a selective endothelial barrier that restricts molecular transport from the retinal microvasculature, located in between neuronal nuclear layers, to neighboring neural tissue [1]. Likewise, the blood–brain barrier (BBB) controls molecular transport to brain neural tissue. These blood–neural barriers are composed of perivascular mural and glial cells that share a common origin in the central nervous system (CNS) vascular development [2,3], form the retinal and brain neurovascular units (NVUs) and play multiple roles in barrier stabilization and maintenance to protect against neurotoxicity [4,5,6]. Additionally, both neurovascular barriers have similar cellular properties [7], including the formation of specialized intercellular junctions and downregulation of transcytosis [8,9], which limit paracellular and transcellular permeability, respectively. Microvascular diseases linked to chronic or genetic conditions often lead to barrier breakdown and neurodegeneration in the retina or brain. iBRB dysfunction and vascular leakage are associated with some of the most predominant ocular diseases such as diabetic retinopathy (DR) and age-related macular degeneration (AMD) [10,11]. BBB breakdown was suggested to be involved in the pathophysiology of several neurological disorders such as Alzheimer’s disease, Parkinson’s disease, Huntington’s disease, amyotrophic lateral sclerosis and multiple sclerosis [12]. However, the mechanisms driving barrier breakdown and their link to pathophysiology of neurodegenerative diseases are not yet fully characterized [13]. Pre-clinical studies in animal models poorly translate to humans and represent a major hurdle in the drug discovery pipeline, since 80% of potential treatments fail in clinical trials [14]. Therefore, there is a need for human-based models that faithfully recapitulate the pathophysiology of human diseases, accelerate drug discovery processes, and better predict patient response to treatments.

Human advanced in vitro models are becoming widespread tools to study cellular interactions and function in many organ-specific contexts [15]. This is made possible through technological advances in bioengineered microphysiological systems (MPS) that support three-dimensional (3D) cellular self-organization [16], such as microfluidics or 3D patterning [17,18] in combination with biocompatible hydrogels [19,20]. While these individual technological advances have progressed incrementally over decades, their combination was only achieved recently. Resulting MPS now facilitate the formation, maturation and maintenance of complex organotypic structures in well-defined and customizable microenvironments [18,21,22]. Human-based MPS have already been developed for tissues such as lung, gut, brain and tumors, yet few integrative models exist for ocular tissues and iBRB modeling [23,24]. Considering the implications of barrier breakdown and vascular leakiness in ocular and neurodegenerative diseases, there is an urgent need to elucidate the pathophysiological mechanisms of vascular instability and accelerate therapeutic interventions.

MPS are especially important to model the NVU, where close interactions between endothelial cells (ECs), pericytes and astrocytes are crucial for proper function [24,25,26]. Various in vitro models and assays are being developed to investigate BBB function and pathogenesis. ECs in mono-culture or in co-culture together with pericytes and astrocytes have been grown in different configurations as 2D monolayers and more recently in 3D MPS to interrogate barrier properties in standard culture or in response to treatments disrupting or maintaining vascular permeability [27,28]. Only recently, the assembly of endothelial and perivascular cells was accomplished in a manner that resembles NVUs found in CNS micro-vasculatures, by self-organization of cells into microvascular networks (MVNs) in MPS [29]. Deeper characterization of vascular models using primary cell sources alone, or in combination with induced pluripotent stem cell (iPSC)-derived ECs, demonstrated acquisition of tissue-specific cellular identities representative of physiological human blood-neural barriers. Furthermore, human-based advanced cellular models have the potential to be customized to specific patient groups/populations and to be scalable for drug discovery purposes. Taken together, these BBB in vitro models showed physiologically relevant morphology and permeability values, and reproduced pathophysiological phenotypes such as vascular leakage and neurodegeneration [30,31,32].

Here, we review similarities in blood–neural barrier and NVU properties in the retina and brain, and focus on recent developments in MPS-based neurovascular barrier models. While bioengineering approaches to model BBB features have already been extensively reviewed [22,25,33,34,35,36,37,38], we propose to harness the strengths of existing MPS-based strategies to generate better iBRB models. Finally, we explore potentials to further personalize cellular inputs and improve pre-clinical testing using MPS.

## 2. Neurovascular Units in Health and Disease

CNS micro-vasculatures contain specialized endothelial and perivascular cells that, together with neurons and glial cells, are organized into NVUs. Well-orchestrated interactions between these different cell groups secure the healthy function of the CNS while dysfunction of any of them can lead to inflammation, neurotoxicity and are associated with neurodegeneration [6,12]. Here, we provide a brief summary of the key components and properties of the iBRB and BBB, before highlighting similarities in their development. We then describe the impact of neurovascular barrier breakdown on pathophysiological outcomes.

### 2.1. Barrier Properties

NVUs consist of specialized endothelial and supporting pericytes and astrocytes that closely interact to establish and maintain barrier properties. Increasing evidence suggests that ECs in NVUs possess tissue-specific characteristics that distinguish them from peripheral ECs and contribute to barrier properties. First, retinal and brain ECs express specific tight junction proteins, such as Claudin-5 [39] and Occludin [40], that are indispensable to restrict paracellular permeability [41]. Second, transcellular transport across the retinal and brain endothelium is lower compared to peripheral endothelia [42,43]. This is partly due to high expression of Mfsd2a, a lipid transporter that inhibits transcytosis in retinal and brain ECs [8]. In addition, intracellular regulators of transcytosis described in epithelial cells (e.g., Rab25 and Rab17) are absent in CNS-ECs [5]. Third, a specific set of transporters regulates the influx and efflux of substances. For example, ECs at the iBRB and BBB have high expression of the glucose transporter GLUT-1 [44] and the efflux pump P-gp [45]. Finally, the retinal and brain endothelia express low levels of leukocyte adhesion molecules (LAMs) such as ICAM-1 and VCAM-1 [46], which restricts the migration of peripheral immune cells across the healthy iBRB and BBB.

Surrounding and closely interacting with ECs are pericytes and astrocytes [47,48]. Pericytes line the micro-vessels, share and contribute to the basement membrane of those vessels and regulate endothelial proliferation [49]. The ratio of ECs to pericytes in the retina and brain is approximately 1:1–3:1, whereas this ratio is much lower in the lung (10:1) and skeletal muscle (100:1) [50,51]. Interactions between pericytes and ECs induce TGF-β activation, which increases basement membrane synthesis and reduces its degradation via proteinase inhibitors such as TIMP-1 [51,52]. During angiogenesis, PDGF-B secreted by sprouting ECs binds to heparin sulfate proteoglycans of the extracellular matrix (ECM) and forms a concentration gradient that leads to the recruitment and attachment of PDGFRβ-expressing pericytes, which promotes vascular stabilization and maturation, and plays a role in controlling vascular permeability [3].

Astrocytes support iBRB and BBB integrity through end-feet projections, which are highly specialized and polarized structures that cover almost the entire abluminal surface of the blood vessels [48]. Astrocytes secrete Sonic hedgehog (Shh) that increases tight junction protein expression in ECs and reduces expression of leukocyte adhesion molecules [53,54]. Co-cultures of ECs with astrocytes also increased tight junction expression in ECs and reduced paracellular permeability to sucrose [55].

Important and often neglected components of the NVU are the surrounding extracellular matrix molecules of the glycocalyx and basement membrane, located at the luminal and abluminal side, respectively. The basement membrane is composed of collagen IV, laminin, fibronectin, nidogen and perlecan [56], with which the cells interact primarily via integrins. Proteoglycans and glycoproteins of the glycocalyx are also involved in mediating cellular signaling. Astrocyte-derived laminin, for example, promotes BBB integrity by increasing EC tight junction proteins and regulating pericyte maturation, and conditional knock-out of laminin gamma-1 in mice leads to BBB breakdown [57].

### 2.2. Neurovascular Development

Blood vessels in the brain originate from the perineural vascular plexus (PNVP). In humans, around 5 weeks after conception, endothelial sprouts from the PNVP invade the CNS and start to form a vascular network [58] (Figure 1a). In parallel, the retina is vascularized from the optic nerve outward [4] (Figure 1b). During this angiogenic process, signaling pathways that drive angiogenesis in peripheral tissues are activated. These include VEGF and Notch signaling, while PDGF-B/PDGFRβ [52] and ANG/TIE2 [51] are involved in perivascular cell recruitment. In contrast, the Wnt signaling pathway is activated specifically during brain angiogenesis but not in other vascular beds, and controls expression of CLDN5 and PLVAP [59,60]. Early signs of blood–CNS barrier formation are observed, with new vessels exhibiting barrier properties such as the expression of proteins involved in the formation of specialized intercellular junctions (e.g., CLDN5), expression of efflux transporters (e.g., MDR1/ABCB1), and reduced transcellular transport [42,61,62].

Important signal transduction pathways involving ECs and pericytes at the NVU are PDGF-B/PDGFRβ, ANG/TIE2 and TGF-β/TGFβR2. The PDGF-B/PDGFRβ axis is indispensable in pericyte recruitment and BBB stabilization as well as astrocyte end-feet polarization, as demonstrated in pericyte-deficient mouse models [63,64]. Ang1 in both iBRB and BBB is thought to originate from pericytes and astrocytes [65,66], although some studies argue that Ang1 is not produced by pericytes in retinal vessels, and that Tie2 may be activated by blood flow-mediated shear stress instead [2,67,68,69]. Finally, TGF-β signaling is multifaceted, affecting endothelial and pericyte proliferation and barrier integrity [41,70,71,72].

Astrocyte networks populate the developing retina and form a network to guide the retinal vascularization process by radial outgrowth from the optic nerve to the periphery [73] (Figure 1b). In the brain, astrocyte development and maturation take place late in embryonic development and continue after birth [74]. Since developing vessels in the retina are leaky despite the presence of astrocytes [62], and astrocytes are not present in the early stages of brain development [75], they are not likely play a key role in initial barrier formation, but rather in barrier maintenance through secreted factors [57,76,77].

### 2.3. Pathophysiology

Disruption of iBRB/BBB promoting signaling pathways, progressive loss of vascular cells and dissolution of endothelial junctions result in barrier breakdown and neurotoxicity (Figure 1c).

**Diabetic retinopathy.** In diabetic conditions, different mechanisms were identified to cause pericyte loss and other diabetic retinopathy-(DR)-associated changes. Hyperglycemia induces pericyte-specific activation of PKC-δ and SH-1, leading to PDGFR-β dephosphorylation and resistance to PDGF-B secreted by ECs, and resulting in pericyte apoptosis [78]. In addition, retinal capillary pericytes were found to accumulate vasculo-toxic molecules in diabetic conditions, leading to their detachment from ECs or selective pericyte degeneration [79]. For instance, soluble epoxide hydrolase (sEH) expression and accumulation of 19,20-dihydroxydocosapentaenoic acid prevented the formation of N-cadherin and VE-cadherin adherens junctions, thus disrupting pericyte-EC and inter-EC junctions [80]. How these different mechanisms of pericyte loss are connected remains unclear, but sEH inhibition was suggested to increase PDGF-B expression thereby restoring PDGF-mediated survival actions, besides rescuing the formation of adherens junctions and preventing the dissociation of vascular cells. Similarly, Notch3 expression in pericytes promotes survival and EC-pericyte interactions through expression of N-cadherin, so interruption of Notch signaling can result in vascular instability [81,82]. Activation of the Notch pathway was also shown to modulate PDGFRβ signaling independently of PDGF-B. Moreover, pericytes play a direct role in regulating the iBRB by influencing gene expression patterns in ECs and polarization of astrocyte end-feet [64]. Pericyte loss thus leads to losing astrocyte-derived components secreted from the end-feet. In sum, mechanisms causing pericyte loss in DR interfere with paracrine signaling axes between pericytes and ECs such as PDGF-B/PDGFR-β, or with junctions binding pericytes and ECs such as N-Cadherin and Notch3.

**Figure 1 jpm-12-00148-f001:**
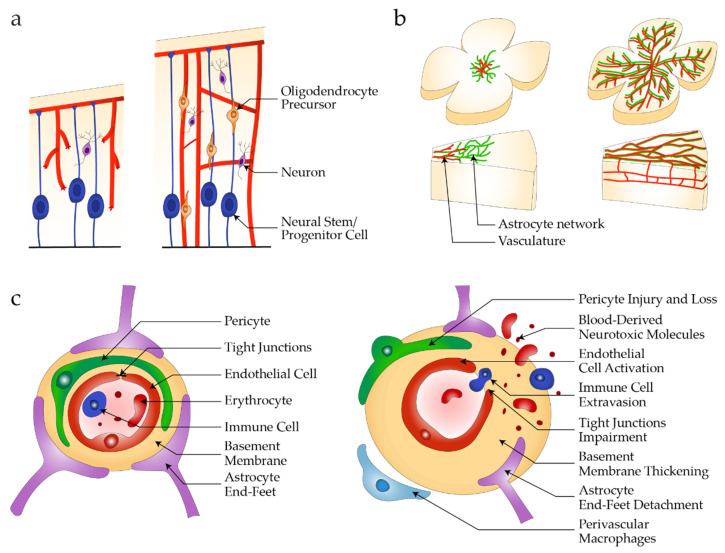
Healthy and diseased NVU. (**a**) Vascularization in brain development originates from the PNVP with sprouting vessels extending along neural stem and progenitor cells and oligodendrocyte precursors that give rise to differentiated neuronal cell types; (**b**) In retinal development, an astrocyte network forms radially from the optic nerve to the periphery followed by sprouting vessels that vascularize the neuroretina following the astrocyte network, as shown in the retinal flat mounts and cross section of the retinal layers; (**c**) Neurovascular unit composed of endothelial cells, pericytes and astrocytes with structural features in healthy conditions (**left**), and during barrier breakdown (**right**). Adapted from [73,83].

**Neurodegenerative diseases.** Neuronal pathologies such as Alzheimer’s disease (AD), Parkinson’s disease, Huntington’s disease and amyotrophic lateral sclerosis are highly associated with compromised BBB integrity. Specifically, in AD, BBB breakdown is correlated with amyloid-β (Aβ) protein accumulation on the vascular walls and the reduction in pericyte coverage [12,84,85]. Interestingly, increased BBB permeability, which can be measured in vivo using several techniques (PET, MRI, near-infrared time-domain optical imaging) [86] correlates well with cognitive decline in humans [87,88,89], and may actually precede it. Furthermore, the expression of transporters, tight junction proteins, and membrane transport regulators is also altered in AD [12,90], suggesting that multiple aspects of BBB function are affected during the course of the disease. Whether these changes have a causative role on the pathophysiology of disease or are instead a secondary result of neurotoxicity and inflammation in the brain is still being investigated.

**Monogenic diseases.** Given the importance of signaling pathways on blood vessel formation and maintenance, as well as on BBB integrity, the improper function of a single protein can be expected to result in devastating outcomes. Indeed, a large number of human inherited monogenic diseases have been identified as disrupting BBB function [65,85]. Typical examples include mutations in *PDGFB* or *PDGFRB* growth factor/receptor causing primary familial brain calcification (idiopathic basal ganglia calcification, Fah’s disease), mutations in receptor *NOTCH3* associated with cerebral autosomal-dominant arteriopathy with subcortical infarcts and leukoencephalopathy (CADASIL) and tight junction *OCLN* in band-like calcification with simplified gyration and polymicrogyria (BLC-PMG), and mutations in basement membrane *COL4A1* and *COL4A2* underlying cerebral small vessel disease or *LAMA2* in congenital muscular dystrophy 1A (MDC1A). The resulting dysfunctional protein often disrupts normal function in specific cell types, including pericytes, vascular smooth muscle cells and ECs. Deeper study of these genes vulnerable to mutations might provide new insights into disease-related vascular dysfunction and potential new therapeutic strategies.

## 3. Strategies to Mimic NVU Architecture and Function

The retinal and brain NVU share functional characteristics and develop in parallel, and the retinal in vivo model has been extensively used to study vessel formation and maturation [62]. In contrast, in vitro models with human cells have largely focused on recapitulating features of the brain NVU. Here, we present both retinal and brain barrier models with applications in drug development, and suggest that approaches developed in connection with BBB models might also be applied to iBRB modeling, despite some region-specific signaling differences [91].

### 3.1. Preliminary Considerations

**Cellular sources.** In the development of BBB models, one of the major challenges is to maintain consistency in critical functional parameters such as permeability. In this regard, the cell source has been identified as a major cause of variability since cells obtained from different donors can show substantially different behaviors in culture, such as self-organization capacity, response to treatment and/or barrier properties. Models of the iBRB or BBB have been developed with ECs, and also with pericytes, astrocytes and neurons obtained from primary tissue or differentiated from iPSC lines [24,25]. In order to recapitulate in vitro the key properties of CNS micro-vasculatures, tissue-specific features, such as higher expression of tight junction proteins and transporter receptors, are required for input cells.

Human primary vascular and perivascular cells obtained from donors have been widely used in different endothelial barrier culture models, and while primary ECs express high levels of barrier markers, the nature of the model system (2D or 3D) and the other cell types used (e.g., pericytes and astrocytes) can dramatically influence expression levels, and as a result, the models only partially recapitulate physiological barrier function, as shown by permeability assays [25,92]. Moreover, their barrier properties vary and are limited by cell passage number and donor-associated heterogeneity [93]. In particular, human umbilical vein endothelial cells (HUVECs) have been reliably used as a source for vascular models, and in the presence of supporting human lung fibroblasts showed effective self-organization into microvascular networks with low permeability values [94,95,96,97].

Human iPSCs are becoming a preferred cell source as differentiation protocols are being improved and cellular identity and maturity are becoming better characterized [98]. Donor-specific iPSCs are used to generate multiple cell types in the same genetic background, and reconstitute complex in vitro models such as the NVU, corresponding to individual patients. Various protocols are available to induce ECs and other components of the NVU [99]. Direct conversion of induced brain microvascular endothelial-like cells (iBECs) were previously described [100], yet recent work showed that the identity of these cells do not correspond to ECs but are instead neuroepithelial cells [101]. These findings raise questions on the applicability of these cells for modeling BBB-specific processes such as immune cell migration or transcytosis. On the other hand, in a recent publication, data were presented showing that primary brain ECs and iPSC-derived ECs share similar gene expression levels of key genes involved in the regulation of BBB function when they are in tri-culture 3D systems [28]. Primary cells and iPSCs are both promising tools to generate in vitro models suitable for physiological and pharmaceutical studies of the iBRB and BBB (Figure 2). For this, cell lines need to be validated for the expression of relevant markers, response to angiogenic stimuli, low permeability values, and capacity to form microvascular networks. In this regard, authors should strive to analyze and report the same genes measured in previous studies and use comparable metrics for function in order to facilitate direct comparisons.

**Gene Editing.** Alongside the development of organotypic in vitro systems, the relevance of disease models can be further enhanced by genetic manipulation, to investigate target mechanisms or pathogenic mutations associated with diseases such as retinal dystrophies. Taking into consideration cell culture requirements when making genetic modifications, including the selection, characterization and expansion of stable cell lines, immortalized cell lines are better suited than freshly isolated primary cells or iPSC-derived cells that have a very limited proliferation capacity. Ideally, genetic modification and stable cell line generation are carried out in a suitable parent iPSC line before differentiation. Nevertheless, genetic modification and associated experimental steps of clonal selection could impact the differentiation efficiency of iPSC lines and downstream phenotyping [102]. Therefore, it is important to test in parallel several iPSC lines or, if unavailable, iPSC clones to distinguish signal from noise during assay measurements.

**In vitro modeling.** Significant advances have been achieved in the development of in vitro BBB models. A variety of models has indeed been referenced and can serve different applications in particular with the engineering of drugs/compounds that could (i) overcome the BBB restricted permeability and target the brain parenchymal cells such as neurons, glial cells or even brain stem cells, but also metastatic brain tumors or (ii) protect from and repair BBB lesions or dysfunctions. In addition to their potential for predictive drug studies and therapeutic applications, such models represent an important asset for dissecting the complex interplay between the different NVU components in steady-state homeostatic conditions as well as in disease, which is often characterized by a primary defect in BBB permeability and protective properties. Indeed, by combining co- and tri- cultures of relevant cell types in a gel matrix, well-calibrated platforms can be designed to better recapitulate the physiological function of the human microvasculature at the capillary level. Since numerous neuronal and vascular diseases have been linked to disrupted neurovascular interactions and BBB failure, multi-component in vitro systems, although challenging to validate, represent promising approaches in tackling these ever increasing health issues.

Current efforts to evaluate transport of biologics to the CNS rely primarily on in vivo studies with rodents. However, recent evidence shows clear differences in the molecular composition of the human BBB compared to mice [103]. In addition, animal studies are not amenable to extensive screening. Therefore, human in vitro models of the BBB can support drug development by improving confidence in targets, studying drug mechanism of action and/or enabling lead optimization campaigns.

Progress in BBB modeling is also likely to benefit related research into the development of iBRB models in health and disease. Further, these developments could provide a rationale for engineering a more complex multicellular system that would include both the inner and the outer components of the BRB for better functional integration.

### 3.2. 2D Models

**2D layers.** Historically, 2D models exemplified with Transwell systems were the first approaches aiming to recreate endothelial barrier properties in vitro. In these settings, ECs placed on top of a porous flat membrane, in mono- or co-culture with pericytes, and with supporting cells such as pericytes and/or astrocytes grown on the other side of the membrane or at the bottom of the well, form monolayers. These multi-component systems enable direct measurement of permeability, but also the spatiotemporal behavior of (i) cell migration through the cell layers and (ii) dynamic interactions between the different cell types of the reconstituted platforms [104,105,106,107,108,109,110,111,112]. A major advantage of these systems is the direct electrical assessment of the trans-endothelial electrical resistance (TEER) and solute values for permeability [113]. Additionally, the use of each side of the membrane support, which can be controlled in its composition, porosity and thickness in conjunction with advanced transcriptomic analysis, provide simplified but informative 2D experimental frameworks [114,115]. Monolayers thus provided a first approach to investigation of the barrier properties of the iBRB and BBB using various readouts.

**2D organs-on-a-chip.** To further improve the relevance of these widely used systems and recreate the tissue and organ complexity within a dynamic microenvironment with physiological flows, 2D organ-on-a-chip models were introduced by seeding different cell types inside a microfluidic device under more controlled conditions that would be fully compatible with cell growth and critical multi-lineage network development. The addition of media flow and control of shear stresses provide a more realistic biomechanical framework which can still be combined with TEER measurement while allowing high-resolution live imaging [116,117]. Various 2D systems have managed to partially recapitulate iBRB or BBB functions such as permeability and high expression of tight junction proteins [30,31,32,117]. Despite the 2D organ-on-a-chip “biomimetic” models offering several advantages over the 2D monolayer models, they are still limited in their morphological relevance, in particular because of the large size of their vasculature-like structures, which largely exceeds the human retinal or brain microvasculature. 2D organs-on-a-chip nonetheless improve barrier properties and increase the co-culture possibilities by introducing additional cellular compartments, which could be applied to modeling the iBRB with additional cell types, such as neurons, in different configurations.

### 3.3. 3D Models

**Blood****-brain barrier organoids.** Instead of co-culturing NVU cell types spatially separated on a trans-well, Urich et al. noticed that when primary human brain ECs, pericytes and astrocytes are mixed in hanging drop cultures, they spontaneously self-assemble into organoids [118]. The organoid core mostly contains astrocytes, while pericytes wrap around the core and the ECs form a continuous layer on the organoid surface. Interestingly, ECs and pericytes also form spheroids when cultured individually, while astrocytes alone or together with ECs are not able to form clusters. Co-culturing ECs, pericytes and astrocytes in organoids also upregulated surface VE-Cadherin, CD31, and P-gp [119] while ICAM-1 and -2 were downregulated. This change in endothelial adherens junction molecules, efflux pump and leucocyte adhesion molecule (LAM) expression was not found in a trans-well setup [118,119], which is likely due to the lack of, or very limited, direct cellular contact in trans-wells.

Various methods have since been used for BBB organoid formation, from hanging drops [118,120,121] to agarose [119], ultra-low attachment plates [122], micro-patterned hydrogels [123] and custom-made microfluidic spheroid arrays [124]. Cho et al. assessed the BBB tightness and functionality by demonstrating P-gp activity and LRP-1 mediated transcytosis of Angiopep-2 into the core of a tri-culture organoid made with primary human brain microvascular ECs (BMECs), pericytes and astrocytes. To demonstrate the aptness of the BBB organoid model for drug screening purposes, they evaluated barrier penetration of fluorescently labeled peptides, and verified their ability to cross the BBB into the brain parenchyma in a mouse model. Simonneau et al. scaled up the BBB organoid assay and used CRISPR/Cas9 edited ECs to identify critical regulators of endocytosis across the BBB model. Using this model, they demonstrated that clathrin, but not caveolin, is required for transferrin receptor-dependent transcytosis of antibody shuttles.

Others have focused on increasing the cellular complexity of BBB organoid models. Nzou et al. added oligodendrocyte progenitors, microglia and neuronal progenitors to the organoid core, all of which were iPSC-derived. They demonstrated selective neurotoxicity by exposing the organoid to 1-methyl-4-phenyl-1,2,3,6-tetrahydropyridine (MPTP) and 1-methyl-4-phenylpyridinium (MPP+). The neurotoxin MPP+ is hydrophilic and thus not able to penetrate an intact BBB, while MPTP can cross the barrier and become metabolized to MPP+ in the presence of glial cells. Nzou et al. demonstrated these different cytotoxic effects in BBB organoids [120]. Instead of adding iPSC-derived cells, Kumarasamy & Sosnik combined primary human BMECs, pericytes and astrocytes with rat cortical neurons and microglia, and used this model to screen for nanoparticle penetration [125].

The overall advantages of BBB organoid models are their ease of assembly and culture, cost-effectiveness and scalability. They can be used for screening small molecule BBB penetration [119] and large molecule transcytosis [123]. By combining this model with genetic engineering on selected cell types [123], BBB organoids can also help to investigate mechanisms of transcytosis. While no similar model exists using retinal microvascular cells, this approach could be applied to model and compare transport mechanisms across the iBRB with medium to high throughput.

**3D tubular channels.** The constant need for better recapitulating organ formation and their vasculature led to a shift of current microfabrication technologies to 3D systems where cells are embedded in hydrogels. The development of 3D-like vascular systems, together with co-culturing different cell types adjacent to each other in hydrogel scaffolds, takes a step closer to mimicking the in vivo complexity of the vasculature. More specifically, microfluidic channels are coated with a layer of ECs to form a tube-like structure, which is adjacent to a gel mixed with different cells of interest. For modeling the BBB, astrocytes and neuronal cells are embedded in adjacent gels to establish a BBB-like organization that can have various applications such as disease modeling [126,127,128]. For the iBRB, an endothelial barrier was used to probe permeability in response to disease triggers [129]. Similar models were achieved using the viscous fingering method or by casting the gel around needles or wires that can subsequently be removed leaving hollow tubular channels, all within a petri dish or microfluidic device, which can subsequently be seeded with cells of interest [130,131,132]. While these models improved BBB properties such as permeability values, morphology and tight junction proteins expression levels, they fail to fully recapitulate the morphology (diameter, branching pattern, etc.) and permeability of the BBB observed in vivo. In particular, the size of the lumens are much larger than in vivo microvasculature and permeability values generally tend to be one to two orders of magnitude higher than in vivo measurements [25].

**3D microvascular networks.** 3D self-assembled MVNs are a recent approach used to recapitulate features of the human brain vasculature more precisely. In this approach, cells of interest are mixed with a hydrogel and acquire BBB-like structure and function following self-assembly. Several published models showed improved relevance for mimicking the human system in terms of vasculature morphology and barrier properties in comparison with the previously described 2D and 3D in vitro models [29,133,134] and better recapitulated pathophysiological features of neurodegenerative diseases than in other in vitro models. For example, in Alzheimer’s disease, pericytes showed a detrimental role in APOE4-mediated amyloid accumulation in the cerebral vasculature [135], and in cancer astrocytes secreting the C-C motif chemokine ligand 2 promoted cancer cell transmigration in the BBB [136]. Advanced transcriptomics and proteomics analysis can be readily obtained after cell extraction from the devices [28]. Barrier permeability can be measured by perfusing different sizes of fluorescent tracers or compounds of interest tagged with a fluorescent marker and measuring accumulated signal in the extra-vascular space [25,27,97]. Overall, direct comparison between a 2D co-culture trans-well system and a 3D co-culture vascular network, all using the same cells, show uniformly lower permeabilities with the 3D network, and these values are comparable to in vivo permeability measurements [27].

Introduction of media flow in a continuous circulating direction was described in similar models [137], but most 3D self-assembled BBB MVNs have not been tested yet under flow conditions. It would be beneficial for these systems to be continuously perfused with media flow using circulating pumps since the shear stresses acting on the vascular walls are known to decrease permeability values, in comparison to static models, and to enhance barrier function [137,138,139,140]. Furthermore, applying 3D self-assembly to recapitulate the iBRB would allow for better models with a microvasculature-like morphology, multicellular organization of co-cultured cell types with direct cellular interactions, and enable a more accurate representation of disease phenotypes.

**Vascularized brain organoids.** While there are several cortical organoids for development and disease modeling that have formed features of brain tissue [141,142], there is a need to vascularize these systems to ensure proper growth and better functionality. BBB organoids have been developed and grafted onto mice for perfusion. They mimic in vivo BBB properties such as high expression levels of tight junctions and adherens junctions, low permeability of different molecules and expression of efflux transporters [143,144,145]. Vascularization of brain organoids has recently benefitted by the incorporation of human embryonic stem cells in the organoid that ectopically express the human ETS variant 2 (ETV2), but it is unclear to what extent these ECs acquire a functional BBB identity [146]. However, there are challenges that the field needs to overcome by improving the organoid formation and culture methods and by increasing their reproducibility. Similarly, retinal organoids have been differentiated but lack vascularization, and thus tissue-specific iBRB properties, for further interrogation [147,148]. This remains a promising direction for increasing the cellular diversity and complexity of the in vitro models to an anatomical level.

Overall, each approach has strengths but also limitations (Table 1). Research labs can combine these approaches depending on the readouts they are interested in. Current tools can provide morphological and functional characteristics of the iBRB/BBB but further improvement is still necessary. Additional cellular and tissue components can be added to the systems such as lymphatic vasculature and microglia that have been shown to be important components of the NVU and the retina or brain in general. In sum, the majority of novel human-based in vitro model developments focus on the BBB while fewer iBRB models are being established. Given the physiological similarities between the iBRB and BBB, we propose that many of the novel approaches developed for the BBB can be applied to accelerate model development for the iBRB. Indeed, retinal and brain ECs share similar CNS vasculature-specific markers, such as Claudin-5, that make assay parallelization possible for this type of readout [41]. Likewise, pericytes and astrocytes are organized similarly in the retinal and brain NVU, so that in vitro iBRB and BBB platform designs could be used interchangeably depending on the degree of complexity and the variety of cellular components [81]. Generally, it has been shown that iPSC-derived ECs tend to acquire a CNS-like identity when cultured in the corresponding BBB-specific niche [28,29], which could also be potentially applicable to the iBRB. However, it is unclear if tissue-specific primary cells could be used interchangeably between iBRB and BBB models since their cellular identities despite being similar are not identical, and the acquisition of tissue-specific gene expression patterns has not been demonstrated. Finally, the iBRB and BBB have additional tissue-specific cell types, such as Müller glia and neurons that would only be introduced in their respective model.

## 4. Outlook

A significant number of drugs fail during Phase II, during which the efficacy of a given molecule is evaluated in a disease population [149]. This is due to inadequate prediction of human efficacy during preclinical development. Therefore, developing methods that better capture drug performance in patients may help to bring more drugs to clinical approval. In vitro human models that recapitulate critical aspects of human biology and disease are one class of tool that can screen therapeutics early in the development process and inform decisions as to whether to move forward with a molecule, or stop its development if the selection criteria are not met. Information gathered from human models can span the investigation of drug mechanism of action, prediction of on- and off-target effects, and evaluation of transport properties or safety. Altogether these data will improve compound optimization, clinical candidate selection and human dose prediction accuracy before clinical trials in humans.

More specifically, microphysiological neurovascular barrier models that contain human NVU components and capture specific NVU function (e.g., permeability, transport) can provide a better understanding of human barrier biology and interrogate barrier dysfunction in disease. When the pathophysiology of the disease is well-documented, as for DR, it is easier to validate the model by the observation of critical pathological events (e.g., pericyte loss, basement membrane thickening, vascular leakage). While in vitro models of the NVU comprise a powerful tool to understand disease processes and screen potential therapeutics, major challenges for broad implementation remain. These include reproducibility, scalability and utility of the readouts, matching the complexity of the model for the specific question, ease of use and throughput.

### 4.1. Readouts

In vitro iBRB/BBB models are well-suited to measure barrier permeability and transport of molecules. These readouts can be used both for compound ranking and selection, as well as for investigating pathways and molecular mechanisms. Barrier permeability can be assessed by TEER or fluorescently labeled molecules such as dextrans in combination with time-lapse imaging, depending on the model and platform. Permeability readouts can be used to assess whether barrier integrity is affected by disease triggers (e.g., VEGFA-induced permeability [129]) and test compound efficacy to prevent and/or restore barrier dysfunction. As previously discussed, crossing of molecules at the BBB can be quantified by adding a fluorescent tag to the compound of interest and measuring the accumulation of signal in the perivascular compartment (or the organoid core in case of iBRB/BBB spheroids [119,123]). This measurement can help optimizing compound properties, or studying mechanisms of receptor-mediated transcytosis [123]. In addition, more complex models that contain cell types of the parenchyma can be used to assess which cell types a molecule of interest targets after crossing the iBRB/BBB, and whether the molecule of interest is effective on its target (i.e., pharmacodynamics) or also affects other cell types (i.e., neurotoxicity [120]). Advanced readouts on vascular functionality are also possible with in vitro models containing blood vessels, for example, to measure vascular contractility [150].

### 4.2. Reproducibility

Within in vitro models, the least controllable parameter remains cellular behavior, which originates and differs from various unpredictable factors. Batch-to-batch variability is common when using biological materials. In the case of primary cell lines and iPSCs, differences in donor profile (age, sex, genetic background) and isolation procedure (cell sorting, media, quality control) can lead to variable cellular behavior. In addition, for a given cell batch in culture, external factors (e.g., extracellular matrix, serum) can lead to cellular heterogeneity, such as astrocyte reactivity in response to serum in the cell culture medium [151]. Systematic cell phenotyping, validation via expression of specific relevant markers and functional assays can help to control heterogeneity and define standards for the cells to be incorporated in 3D models. In self-assembled micro-vessels or self-organized organoids, variations in cell–cell communication during the assembly process can lead to substantial morphological and functional differences from one batch to another. This variability needs to be accounted for in advanced in vitro iBRB and BBB models and defining relevant functional readouts for each model as quality control parameters will facilitate the implementation and regular use of complex models. Further, the parameters of interest, for example permeability, need to be consistent across experiments. As observed with patient populations, some cells and/or models might respond differently to the same trigger. Classification and stratification of models might help understand disease heterogeneity, response to treatment and develop treatments better suited for a given patient population.

### 4.3. Scalability

The more complex the model is in terms of cell types, matrix and structure, the closer we get to accurately mimic organotypic functions, but the more difficult it is to guarantee reproducibility, ease of use, and throughput. Typically in preclinical development, high-throughput systems such as microtissues/spheroids or organoids in suspension would be used for primary screens while lower throughput systems such as organs-on-chips or assembloids that require more direct handling would be employed for target validation or toxicity assessment. Complex models that include perfusion, additional cell types (e.g., microglia) or self-renewal potential (organoids and assembloids) could also be useful in disease modeling over longer periods of time. While they may provide crucial insights into human biological mechanisms in the health and disease of the iBRB/BBB, they require a high investment in resources and time, and require careful handling.

In contrast, models of intermediate complexity, such as tubular NVUs and self-assembled micro-vessels, have the potential to be improved upon in terms of readout reproducibility. Once this is achieved, they are tools that add value in the drug discovery and target validation process.

### 4.4. Personalization

The promise of personalized medicine revolves around using a patient’s autologous cells or a donor’s allogeneic cells with matching human leukocyte antigens (HLA) for drug discovery or regenerative medicine, to test specific drug response and dosage, or to repair or replace damaged tissue [98]. For this reason, cellular sources, genetic manipulation and the accuracy of downstream phenotyping have the potential to personalize pre-clinical research (Figure 2). Primary cells can be isolated, and iPSCs reprogrammed from healthy and diseased individuals, taking into consideration specific characteristics of population groups such as age, sex, genetic background and predisposition to disease.

In addition, genetic factors are known to influence the onset and progression of ocular diseases such as diabetic retinopathy [152] and age-related macular degeneration [153]. Advances in CRISPR/Cas9 template-mediated gene editing are facilitating the generation of single-nucleotide polymorphisms (SNPs) and single-nucleotide variants (SNVs) in iPSCs [154,155], in order to recreate or rescue associated phenotypes in vitro (Figure 2). Similar strategies have been applied for retinal disease modeling or gene therapy, by knock-in of patient mutations, or correction or knock-out of mutant alleles, respectively [156].

## 5. Conclusions

The retinal and brain micro-vasculatures share specific properties and ensure neuroprotective barrier function in health, and their dysfunction in disease can influence neurodegenerative outcomes. Different retinal and brain micro-vasculature in vitro models have been generated that reassemble native tissue morphology and function, mimicking endothelial barrier properties and multicellular NVU architecture, and recapitulating physiological neurovascular barrier permeability values. MPS are a powerful tool to recapitulate the complex cellular microenvironments and experimental boundary conditions required to obtain relevant multicellular organization and interactions. iBRB MPS models are still lacking and can benefit from development in parallel to new BBB models that leverage this technology. In vitro neurovascular barrier models will allow addressing of the mechanisms of disease initiation and progression causing barrier breakdown, vascular leakage and neurodegeneration in ocular and neurological diseases. MPS-based models have the potential to accelerate pathophysiological insights relevant to human biology and accelerate the development of new therapies, by increasing the controllability over cellular inputs and by matching the level of complexity of a model with its capacity for high-throughput and reproducible testing.

## Figures and Tables

**Figure 2 jpm-12-00148-f002:**
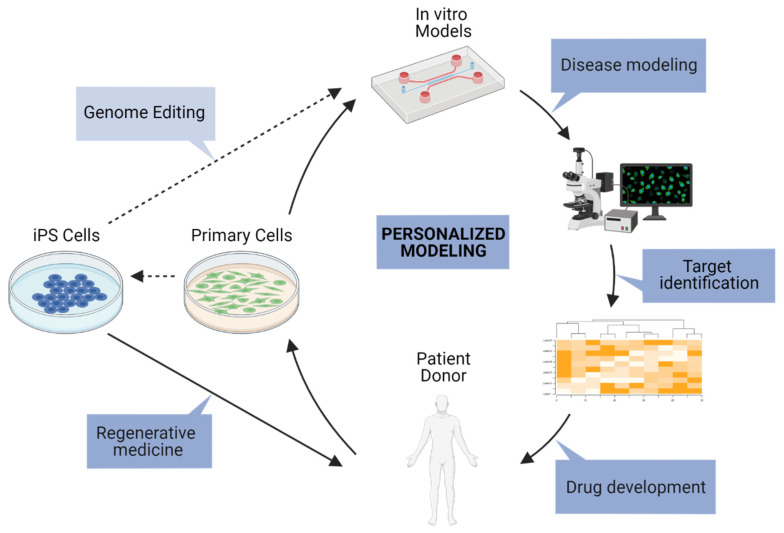
Personalized approaches for neurovascular barrier models. Patient or donor tissue is a source for cellular and genetic manipulation with the aim of treating patients through regenerative medicine or disease modeling approaches. Self-organization of relevant cell types in MPS models provide valuable tools to test targets in a personalized manner and develop drugs that are pre-validated in vitro on autologous cells. Created with BioRender.com.

**Table 1 jpm-12-00148-t001:** Summary of in vitro neurovascular barrier models.

Model	Application	Limitations	iBRB Refs.	BBB Refs.
2D layers (e.g., Trans-well)	Tight junction immunostaining TEER Permeability Imaging Transcriptomic analysis	Stiffness of tissue culture plastic 2D monolayer morphology Tendency for high permeability	[104,105,106,107,108]	[109,110,111,112]
2D organ-on-a-chip (e.g., Emulate)	Tight junction immunostaining TEER Permeability Imaging Transcriptomic analysis	Poor morphology Tendency for high permeability Low throughput	[117]	[30,31,32]
3D organoids	Immunostaining Permeability Transcytosis Imaging medium-high throughput screening	TEER not available Limited cellular self-organization	N/A	[118,119,120,121,122,123]
3D tubular channels (e.g., Mimetas)	Tight junction immunostaining Permeability Imaging	Morphologically large networks Limited cellular self-organization Moderate throughput	[129]	[126,127,128,130,131]
3D self-assembled micro-vasculatures (e.g., AIM Biotech)	Immunostaining Permeability Imaging Transcriptomic analysis	TEER not available Morphological variability	N/A	[28,29,133,134,135,136,137,138,139,140]
		Moderate throughput		

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
