# Peer review of "Microphysiological Neurovascular Barriers to Model the Inner Retinal Microvasculature"

_jpm, 2022, doi:10.3390/jpm12020148_

Round 1

Reviewer 1 Report

A review manuscript by Maurissen et al titled ‘Microphysiological Neurovascular Barriers to Model the Inner Retinal Microvasculature’ describes and compares existing experimental model settings for studies of blood-brain and inner blood-retinal barriers (BBB and iBRB, respectively). The manuscript is well-written and provides a good overview of BBB and iBRB structural organization and existing models to study BBB/iBRB. The manuscript is timely and its focus is of current interest. This work would likely attract attention of a broader audience not limited to researchers who study BBB and/or iBRB but also those engaged in exploration of barrier properties of endothelial and epithelial cells.

However, a few points should be readdressed by the authors.

Major

The manuscript title declares the focus at models of iBRB. In Abstract (line 29), the authors state ‘…few comprehensive models exist for ocular tissues and iBRB modeling...’ and further (lines 32-34) ‘we…propose to leverage MPS-based strategies to develop new iBRB models…’.

However, the authors’ proposal is limited to one sentence (lines 463-465). This idea should be expanded with evidence/rationale that a particular model established for BBB is applicable for iBRB, what should be adjusted, or, otherwise, what the specific hurdles are why this model is not applicable.

In this regard, there are a few ways to improve the manuscript:

  • add notes that would stress the applicability to iBRB studies as a bottom-line in each paragraph in Section 3.
  • expand Table1 with an additional column to segregate references where BBB or iBRB were studied. It would show whether a particular model had been exploited for iBRB research.
  • Expand the phrase ‘…many of the novel approaches…’ (line 464) with evidence/rationale that a particular model established for BBB is applicable for iBRB and what adjustments are need if any.

Minor

  • Reference 28 (Hajal et al) is cited several times. However, I failed to locate the cited work either in Nat. Protoc., Pubmed, or Scopus. Neither title nor author searches gave results. Please, check whether the reference details are correctly indicated, and provide DOI.
  • Check abbreviations are disclosed at the first mentioning. For example, iPSC appears in line 87 but is disclosed in line 259. Notably, some abbreviations (for example, MPP+ or QC) are not disclosed ever.

Author Response

Comments and Suggestions for Authors

A review manuscript by Maurissen et al titled ‘Microphysiological Neurovascular Barriers to Model the Inner Retinal Microvasculature’ describes and compares existing experimental model settings for studies of blood-brain and inner blood-retinal barriers (BBB and iBRB, respectively). The manuscript is well-written and provides a good overview of BBB and iBRB structural organization and existing models to study BBB/iBRB. The manuscript is timely and its focus is of current interest. This work would likely attract attention of a broader audience not limited to researchers who study BBB and/or iBRB but also those engaged in exploration of barrier properties of endothelial and epithelial cells.

We thank the Reviewer for acknowledging the topic of this review on iBRB and BBB models. We will address the major and minor comments below.

However, a few points should be readdressed by the authors.

Major

The manuscript title declares the focus at models of iBRB. In Abstract (line 29), the authors state ‘…few comprehensive models exist for ocular tissues and iBRB modeling...’ and further (lines 32-34) ‘we…propose to leverage MPS-based strategies to develop new iBRB models…’.

However, the authors’ proposal is limited to one sentence (lines 463-465). This idea should be expanded with evidence/rationale that a particular model established for BBB is applicable for iBRB, what should be adjusted, or, otherwise, what the specific hurdles are why this model is not applicable.

We thank the Reviewer for giving us an opportunity to better explain the rationale of our idea that existing BBB models can be leveraged to develop new iBRB models. We modified the text based on the recommendations given to us below by the Reviewer.

In this regard, there are a few ways to improve the manuscript:

  • add notes that would stress the applicability to iBRB studies as a bottom-line in each paragraph in Section 3.

We added clarifications in paragraphs in Section 3 on how each model was relevant and applicable to iBRB studies. The inserted text appears as track-changes in Section 3 at lines 345-346 reading “Monolayers thus provided a first approach to investigate barrier properties of the iBRB and BBB using various readouts.”, lines 359-362 “2D organs-on-a-chip nonetheless improve barrier properties and increase the co-culture possibilities by introducing additional cellular compartments, which could be applied to modeling the iBRB with additional cell types, such as neurons, in different configurations.”, lines 403-405 “While no similar model exists using retinal microvascular cells, this approach could be applied to model and compare transport mechanisms across the iBRB with medium to high throughput.”, lines 451-454 “Furthermore, applying 3D self-assembly to recaptulate the iBRB would allow for better models having a microvasculature-like morphology, multicellular organization of co-cultured cell types with direct cellular interactions, and enable a more accurate representation of disease phenotypes.”, and lines 466-469 “Similarly, retinal organoids have been differentiated but lack vascularization and thus tissue-specific iBRB properties for further interrogation [147], [148]. This remains a prom-ising direction to increase the cellular diversity and complexity of the in vitro models to an anatomical level.”.

  • expand Table1 with an additional column to segregate references where BBB or iBRB were studied. It would show whether a particular model had been exploited for iBRB research.

We added a column in Table 1 to separate references that focused on the iBRB and BBB, showing that no iBRB models were available for 3D organoids and 3D self-assembled microvasculatures.

  • Expand the phrase ‘…many of the novel approaches…’ (line 464) with evidence/rationale that a particular model established for BBB is applicable for iBRB and what adjustments are need if any.

We added a paragraph to highlight the similarities and differences to consider when establishing iBRB models analogous to BBB models. The inserted text appears as track-changes in Section 3 at lines 476-491 reading “In sum, the majority of novel human-based in vitro model developments focus on the BBB while fewer iBRB models are being established. Given the physiological similarities between the iBRB and BBB, we propose that many of the novel approaches developed for the BBB can be applied to accelerate model development for the iBRB. Indeed, retinal and brain ECs share similar CNS vasculature-specific markers, such as Claudin-5, that make assay parallelization possible for this type of readouts [41]. Likewise, pericytes and astro-cytes are organized similarly in the retinal and brain NVU, so that in vitro iBRB and BBB platform designs could be used interchangably depending on the degree of complexity and the variety of cellular components [81]. Generally, it was shown that iPSC-derived ECs tend to acquire a CNS-like identity when cultured in the corresponding BBB-specific niche [28], [29], which could be potentially applicable to the iBRB as well. However, it is unclear if tissue-specific primary cells could be used interchangably between iBRB and BBB models since their cellular identities despite being similar are not identical, and the acquisition of tissue-specific gene expression patterns has not been demonstrated. Finally, the iBRB and BBB have additional tissue-specific cell types, such as Müller glia and neu-rons that would only be introduced in their respective model.”.

Minor

  • Reference 28 (Hajal et al) is cited several times. However, I failed to locate the cited work either in Nat. Protoc., Pubmed, or Scopus. Neither title nor author searches gave results. Please, check whether the reference details are correctly indicated, and provide DOI.

This reference was important for our review and was in press at the time of submission. It is now available online (https://doi.org/10.1038/s41596-021-00635-w).

  • Check abbreviations are disclosed at the first mentioning. For example, iPSC appears in line 87 but is disclosed in line 259. Notably, some abbreviations (for example, MPP+ or QC) are not disclosed ever.

We disclosed abbreviations where indicated by the Reviewer, at other mentionings in the text, and also LAM (line 373).

Reviewer 2 Report

The article by Maurissen et al. have reviewed various advanced cellular models for studying the blood retina barrier (BRB). They have compared these BRB models to blood brain barrier models owing to similarities between the two systems. This articles provides a comprehensive review of the challenges associated with the development of such model systems and how development of these systems would be useful for understanding and developing therapies for diseases such as diabetic retinopathy where BRB is compromised. 

The review article looks exhaustive and no changes are required.

Author Response

Comments and Suggestions for Authors

The article by Maurissen et al. have reviewed various advanced cellular models for studying the blood retina barrier (BRB). They have compared these BRB models to blood brain barrier models owing to similarities between the two systems. This articles provides a comprehensive review of the challenges associated with the development of such model systems and how development of these systems would be useful for understanding and developing therapies for diseases such as diabetic retinopathy where BRB is compromised. 

The review article looks exhaustive and no changes are required.

We thank the Reviewer for positive feedback

Reviewer 3 Report

The review is well written with enough info regarding blood retinal barrier, boold brain barrier and MPS-based strategies. I recommend this review for publication.

Author Response

Comments and Suggestions for Authors

The review is well written with enough info regarding blood retinal barrier, blood brain barrier and MPS-based strategies. I recommend this review for publication.

We thank the Reviewer for positive feedback.

Round 2

Reviewer 1 Report

In the revised manuscript by Maurissen et al, the autors have addressed all issues raised in Round 1 review and introduced the requested changes. I have no major concerns regarding the revised manuscript and recommend publication.